# The Role of Information Management for the Sustainable Conservation of Cultural Heritage

Jaione Korro Bañuelos [1], Álvaro Rodríguez Miranda [1,2,*], José Manuel Valle-Melón [1], Ainara Zornoza-Indart [3], Manuel Castellano-Román [4], Roque Angulo-Fornos [4], Francisco Pinto-Puerto [4], Pilar Acosta Ibáñez [5] and Patricia Ferreira-Lopes [5]

[1] GPAC-Built Heritage Research Group, Laboratory for the Geometric Documentation of Heritage, University of the Basque Country (UPV/EHU), Justo Vélez de Elorriaga 1, 01006 Vitoria-Gasteiz, Spain; mirenjaione.korro@ehu.eus (J.K.B.); jm.valle@ehu.eus (J.M.V.-M.)
[2] Department of Applied Mathematics, Faculty of Engineering of Vitoria-Gasteiz, University of the Basque Country (UPV/EHU), 48940 Leioa, Spain
[3] Department of Painting, Faculty of Fine Arts, University of the Basque Country (UPV/EHU), Barrio Sarriena, 48940 Leioa, Spain; ainara.zornoza@ehu.eus
[4] Institute of Architecture and Building Science (IUACC), University of Seville, Reina Mercedes 2, 41012 Seville, Spain; manuelcr@us.es (M.C.-R.); roqueaf@us.es (R.A.-F.); fspp@us.es (F.P.-P.)
[5] Andalusian Institute of Historical Heritage (IAPH), Camino de los Descubrimientos (Carthusian Monastery), 41092 Sevilla, Spain; pilar.acosta@juntadeandalucia.es (P.A.I.); patricia.ferreira@juntadeandalucia.es (P.F.-L.)
* Correspondence: alvaro.rodriguezm@ehu.eus

**Abstract:** Central to the entire discipline of heritage restoration and conservation is the concept of information management. Nevertheless, traditionally, conservation and restoration has been a poorly documented discipline, which has led not only to a lack of standardization and awareness about the processes carried out in the past, but also poses problems both when new restoration works are necessary and for the preventive conservation of the elements of heritage. This study sets out to propose a conceptual framework to explore the relationship between conservation of heritage and information management on the basis of case studies; in particular: a spatial data infrastructure (SDI) of a regional government concerning an endangered plant (wild grapevine) with an important potential for cultural and touristic uses in a wine-making region; an open data guide—the Digital Guide of Andalusian Cultural Heritage; a university repository connected to Europeana, which contains reports and outcomes of projects of geometric documentation of elements of heritage; a repository of an organization in charge of the protection and care of the heritage; and finally, two examples of the use of heritage building information models (HBIM) in complex monuments. After discussing the characteristics of each case, this paper concludes that, although the availability of information and tools is growing, further progress is still necessary concerning the interoperability, outreach and reuse of the different solutions.

**Keywords:** cultural heritage; conservation; information management; digital preservation; valorization of cultural heritage

## 1. Introduction

The aim of this work is to provide an overview of the possibilities regarding the information management related to cultural heritage elements, with special reference to conservation-restoration works. The study selects a set of significant features used to evaluate each option individually in order to contrast them with the necessities and resources of each organization, its user community and the group of heritage elements of which it is in charge. Moreover, some guidelines are proposed for merging different tools in a customized strategy that can meet the expectations and needs of both the organizations and the society concerning the sustainable conservation of cultural heritage.

Cultural Heritage plays a key role in the well-being of both individuals and societies. This premise is clearly stated in the document "Cultural Heritage: a powerful catalyst for the future of Europe" (2020) (Elaborated by European Heritage Alliance, Europa Nostra and Europeana Foundation. https://www.europanostra.org/europe-day-manifesto-cultural-heritage-a-powerful-catalyst-for-the-future-of-europe-just-released/; accessed on 4 January 2021). Indeed, cultural heritage is outlined as follows: (1) it contributes to the attractiveness of the regions; (2) it is the basis for establishing marketing strategies aimed at developing cultural tourism and attracting investments; (3) it is a main agent which creates employment opportunities covering a wide range of fields, from construction related to conservation and maintenance, to a more varied group of small- and medium-sized enterprises (SME), as well as start-ups; (4) it generates substantial fiscal revenues, (5) it is a major source of creativity and innovation; (6) it is a catalyst for a sustainable regeneration; and finally, (7) it is able to offer solutions to the challenges of climate change.

These seven pillars highlight the fact that cultural heritage is a powerful engine for economic growth and a valuable resource for social development; hence its proper conservation—defined by the European standard [1] as the actions taken in order to safeguard cultural heritage in keeping with its historic interest and including the accessibility by current and future generations— becomes essential for achieving these benefits. In the field of conservation ("conservation" encompasses: preventive conservation—measures geared towards avoiding or minimizing the harm, deterioration or loss in the future, usually through indirect actions targeted at the surrounding environment rather than on the elements of heritage themselves; curative conservation—actions on the elements in order to stop or limit the deterioration; and restoration—actions on a stabilized element to improve recognition, understanding and/or use, subject to respecting its cultural value, materials and techniques [2]), documentation is always a requirement because:

- When dealing with curative conservation and restoration, there is a need (before acting) to gather any piece of information that contributes to the knowledge about the object, not only as the previous restorations are concerned, but also regarding the history, meaning, materials, techniques, legal status and so on, in order to decide the most suitable actions to be undertaken in each case [2]; this need becomes especially obvious when dealing with cases in which the documentation does not exist or where it is very scarce (e.g., [3]);
- As regards preventive conservation, careful consideration should be given to both the object itself—including its uses and management conditions—and the surrounding environment to detect conservation problems or risks, and then to enable the design of maintenance protocols, monitoring strategies and emergency response measures that are applied methodically over time in the framework of a continuous improvement [4].

Nevertheless, the key aspects of under-documentation can be noted as follows: professional secrecy, the preconception that conservation and restoration are a simple set of manual abilities, and the lack of resources for both the generation and the management of the information. This has led to the consideration by some professionals that information concerning the objects or monuments was neither necessary for restoration work, nor that said work would add any value to the element of heritage [5,6]. However, three recent changes that imply a rethinking of this situation should be considered:

1. At present, cultural heritage is seen as something dynamic; therefore, the meaning of restoration has shifted from actions that—to a greater or lesser degree—"affect" the heritage to events that contribute to its material history [7]. Likewise, the criteria for restoration have evolved inasmuch as the technologies for the study of the materials and their durability over time have been developed; today, it is known that the products applied in the restoration works (consolidating products, waterproofing materials, biocides, etc.) are not reversible—i.e., they cannot be removed without damaging the material integrity of the object due to infiltration into the porous structure and the changes caused by exposure to aggressive environments (solar radiation, pollution, marine aerosol, etc.). On the contrary, the application of products

modifies the intrinsic characteristics of the materials on which they are applied and conditions subsequent to retreatment;

2.  Restorations are no longer considered as exceptional corrective actions that must go unnoticed for the users. Conversely, restoration actions should be understood within the process of permanent maintenance of the elements of heritage, which has to be known and open to the public [8]. The combination of information from different fields generates knowledge that permits the development of attractive cultural activities [9]. In the same vein, the outreach of the conservation-restoration works is a way of attracting visitors and, hence, generating economic returns. It also benefits the interest and the citizen engagement in safeguarding cultural heritage, the awareness of its meaning, the generation of a collective memory and the promotion of a sense of belonging and identity. Consequently, threats related to inappropriate use and looting are reduced, as well as human action, which is one of the main causes of deterioration;

3.  The third substantial change is the trend to steer the conservation to prevention and the risk management, rather than to the physical action on the elements, through the development and implementation of preventive conservation plans [10,11]; in this way, the deteriorations and losses are avoided without drastic and costly treatments on the objects that could eventually imply a reduction of their durability and authenticity [12]. The identification and evaluation of the risks are based on studies of the current and past situation of the heritage element and its surroundings, by using tools for quantifying the probability of occurrence of each risk, its impact, the extension and the estimation of the resulting loss in value [13,14]. Current methods resort to a series of indexes, such as the vulnerability index, which combines static and structural factors, weather conditions, air quality, urban planning and social agents [15,16]; another method used is the heritage microclimate risk (HMR) index, which defines a level of risk to which the elements are exposed and the relationship with the minimum and maximum values that are defined by standards or according to the historic microclimate, together with the predicted risk of damage (PRD) index, which links the HMR value with the damage risk to a specific object/material that is shown or conserved in a room [17], among others. Moreover, it must also be considered that, owing to climate change and the exponential growth in the deterioration processes of the materials caused by the increasingly aggressive environments, the monitoring and the documentation of these environmental conditions have become strategic for the basic preventive conservation, markedly for cities and coastal areas [18,19]. Finally, information also plays an essential role in dealing with catastrophic events such as natural disasters and armed conflicts, as is illustrated in the disaster risk management (DRM) cycle for cultural heritage [20].

All these changes need efficient information management about the conservation-restoration works in order to be properly implemented, and it is necessary that the information is integrated dynamically with datasets coming from other disciplines, thus making a body of knowledge on heritage elements. Evidently, information by itself is not enough, as supportive computer tools [21] and a sound legal basis are also needed.

The management of the information regarding cultural heritage elements is a very active research area with noteworthy international projects. For instance—mentioning only projects in the H2020 program—the following ones can be cited: eHERITAGE (about virtual reality applications for the outreach of cultural heritage, http://www.eheritage. org/), STORM (management of risks from natural hazards affecting cultural elements, http://www.storm-project.eu/), Iperion CH (a data infrastructure concerning conservation and restoration works, http://www.iperionch.eu/) or InDICEs (measurement of the impact of digitization and the reuse of cultural resources, https://indices-culture.eu/). On the other hand, the experience on cultural heritage is being articulated by means of collaborative knowledge networks, a good example of which is E-RIHS (http://www.e-rihs.eu/); likewise, other programs—such as the COST actions—are also fostering partnerships, as is the case of SEADDA, regarding the preservation of digital datasets from archaeology

(https://www.seadda.eu/). Looking at the future, it is important to highlight that these issues are also in line with the investment strategies of the EU for the following years, which are included in the ambitious program "Horizon Europe" for the period 2021–2027. More specifically, this area of research connects two of the clusters in Pillar 2 "Global Challenges and European Industrial Competitiveness", in particular, the ones entitled "Culture, creativity and inclusive society" and "Digital, industry and space".

In next section, a selection of representative features for the definition of an information management system are identified from the study of recommendations and standards. Next, a review of initiatives related to information management strategies implemented by several organizations concerned with the care and protection of cultural heritage (public administrations, universities, research centers and so on) is carried out, together with the tools available for their implementation. In parallel, a number of case studies are presented in order to explore the applicability and outcomes of each solution. These case studies are then compared using the previously selected set of features. The comparison is followed by a discussion about the benefits, deficiencies, strengths and limitations of each alternative, so that some general criteria about the information management for conservation purposes can be drawn, including interoperability issues, preservation of datasets, outreach strategies and usage monitoring.

## 2. Materials and Methods

There has been a growing number of publications focusing on normalization by way of guidelines and internal procedures of the cultural organizations and standardization bodies. As for the good practices and criteria created by cultural institutions, both international and national organizations should be examined (ICOM, Getty, Historic England, etc.), as well as the experience gathered by means of the national plans for cultural heritage in the different countries [22]. A consideration of key works should include: European Standards EN 16095 and 16096 on the condition reports of the elements of heritage and EN 16853 concerning decision making, planning and implementation, written by the European Committee for Standardization (CEN). Moreover, standards regarding other fields can also apply, such as the rich set regarding the information management (ISO 15489, ISO 30300 . . . ), metadata (ISO 23081), risks management (ISO/IEC 27001, ISO 31000 . . . ), interoperability (ISO 21127, ISO 25964 . . . ) or repositories (ISO 14641, ISO 14721, ISO 17068 . . . ).

With these reference documents in mind, the following eight characteristics were selected as representative of the topics that are considered essential during the definition and the establishment of an information management system: (1) the types of information that are managed by the system (e.g., reports, photographs, 3D models); (2) whether the information stored concerns the element of heritage itself, whether it is about other topics which are also important for the management, or both; (3) the availability of advanced built-in tools for data processing; (4) the possibility of indexing the contents from the outside (for instance, by means of aggregators) so that external systems can provide additional functionalities; (6) the expected profile of the users (experts or general public); (7) whether the management is carried out directly by the organization or it is transferred; and (8) the development costs.

Of note is that, in response to the overview presented in the introduction, different organizations and professionals concerned with cultural heritage feel the necessity of joining forces to design the basis for a diverse and broad documentation, which is also interoperable, enabling it to be useful for a varied group of users. Although all approaches have points in common and there have been many attempts to connect them, significant barriers still remain. This work examines these alternatives considering the type of organizations and backgrounds that have led to the creation of each system (user community, initial needs, type of documents, standards, tools, etc.) and then they are discussed together. For the purpose of this text, three groups were established: "open data" of public administrations—including the spatial data infrastructures (SDI), distributed networks of institutional repositories and databases with advanced capabilities for geographic

and three-dimensional processing. Likewise, these approaches are analyzed by means of selected examples in order to show the variability of the possible situations.

### 2.1. "Open Data" of Public Administrations

Public administrations make great efforts towards transparency of their activities by opening to society the information generated from public funding, it is the open data initiative (an initiative that, in the European context, has a starting point in the Directive 2003/98/EC of the European Parliament and of the Council of 17 November 2003 on the reuse of public sector information (modified by Directive 2013/37/UE), and is continued by successive documents, such as the Commission Recommendation of 24 August 2006 on the digitization and online accessibility of cultural material and digital preservation (2006/585/EC) or the Commission Recommendation of 27 October 2011 on the digitization and online accessibility of cultural material and digital preservation (2011/711/EU), up to the recent Directive (EU) 2019/1024 of the European Parliament and of the Council of 20 June 2019 on open data and the reuse of public sector information).

One of its most prominent instances are the spatial data infrastructures (SDI), i.e., geo-referenced data that, in Europe, are shared according to the Infrastructure for Spatial Information in the European Community (INSPIRE) Directive (Directive 2007/2/EC of the European Parliament and of the Council of 14 March 2007), which offers the possibility of including information about cultural heritage within the theme of "protected sites" [23,24], as well as enabling multitemporal studies [25].

In general, the SDI are oriented to two-dimensional territorial datasets which are to be represented in scales from 1:500 (urban maps) or 1:5000 (as a typical scale for the base maps used for regional planning) to 1 as well as smaller ones (supra-regional and national level studies). This range of scales and the non-consideration of the volumetric nature of the elements make them inappropriate, for instance, for the management of a building for which three-dimensional data at the scales 1:10 to 1:100 are available. However, the SDI include all the coverages of the official maps (roads, rivers, towns, etc.) as well as the information provided by a number of organizations (weather, seismicity, etc.); besides, datasets have topology (e.g., the roads are properly connected and include information about the state of use, maximum speed, etc., so that the way of traveling from one point to another, as well as the time necessary to do it, can be computed). Moreover, the elements can be linked to and from external databases (land registry, national institute of statistics and so on) and, finally, it is important to mention that SDI are rather easy to use, since they can be accessed both directly within the geographic services provided by the different organizations, and indirectly from any geographic information system (GIS) or viewer (e.g., Google Earth®®) on the users' part.

In view of the above, many of the factors that are relevant for the preventive conservation and the promotion of the elements of heritage (such as natural risks, pollution levels, planning regulations, accessibility, location of related cultural assets and so on) can be considered by means of the SDI [26]. In order to see the capabilities of this approach, a SDI of a regional government relating to a natural resource (wild grapevine) with an important potential for enhancing cultural and touristic uses in a wine-making area is presented below (Figure 1) [27]. This SDI was chosen to illustrate two ideas: Firstly, the same risks that may affect cultural heritage (e.g., pollution, uncontrolled urbanization, massification and so forth) are also apparent in other elements. With this in mind, both the monitoring and the evaluation of the damage of those factors can be performed more efficiently. Secondly, the local environment greatly influences the management and uses of the individual elements, for instance, if the attractiveness of the area increases thanks to the SDI, because it provides information for nature excursions related with the wild vines that can be combined with visits to wineries, restaurants, accommodations and other leisure activities. Thus, in managing individual elements of heritage, there needs to be awareness of and coordination with the regional context.

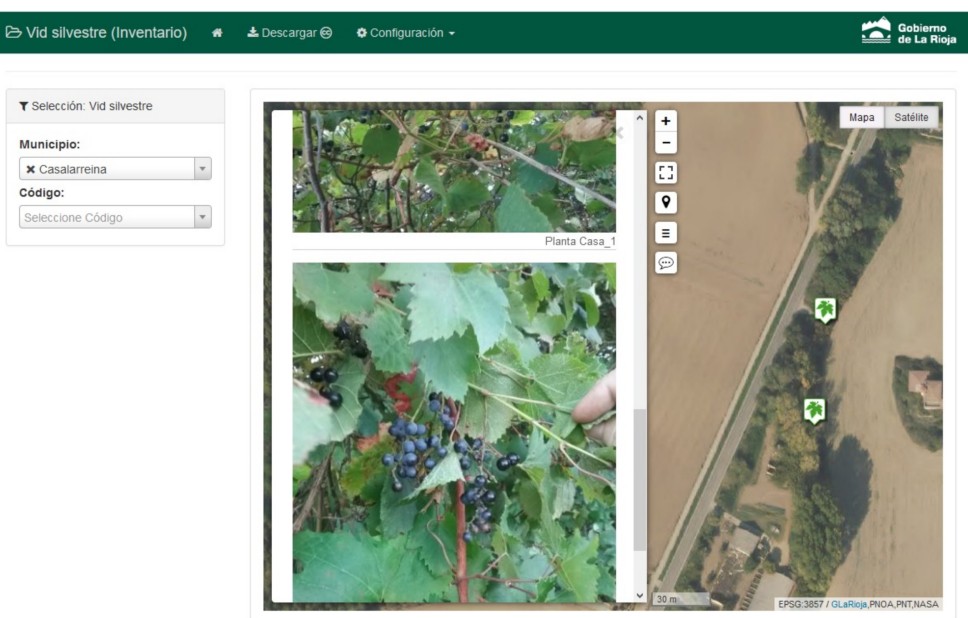

**Figure 1.** Thematic viewer of the regional government of La Rioja (Spain) showing the layer with the location of the specimens of wild grapevine—*Vitis vinifera* L. subesp. *sylvestris* (Gmelin) Hegi—and some of the associated documents (in this case, two photographs) (This thematic map can be seen and used in the viewer of the regional administration: https://www.iderioja.larioja.org/vct/index.php?c=4e38512b51695553786e4143325369447a74484270513d3d and accessed as a geoservice according to the Open Geospatial Consortium (OGC) standards: Web Map Service (WMS) in https://ogc.larioja.org/wms/vidsil/request.php and Web Feature Service (WFS) in https://ogc.larioja.org/wfs/vidsil/request.php) (all web services accessed: 15 February 2021).

The generation of a geospatial service requires two main elements: On the one hand, the information—in this case, the location of the specimens, which was obtained by means of a bibliographical research and a series of field works carried out by experts who identified and registered the significant features and took photographs of each plant. On the other hand, there is an important amount of work concerning the definition of the database, the characteristics of the information uploaded by the providers and the agreements between the different actors involved (the researchers who generate and maintain the information updated, the regional mapping service which manages the infrastructure and the regional government which is responsible for the service) concerning the intellectual and exploitation rights.

In the SDI, both the arrangement of the elements and the way of accessing the information are based on the geographic space. Although the location is always a meaningful feature of heritage, it may sometimes be useful to include a classification regarding the type of heritage (building, site, landscape, etc.) and the possibility to show spatial and associative relationships (e.g., an element of heritage, such as an altarpiece, which is inside another element, for example, a church, buildings that were restored by the same architect and so on). In order to illustrate this possibility, the information system called Guía Digital del Patrimonio Cultural de Andalucía (This record can be accessed through: https://guiadigital.iaph.es/inicio, accessed: 15 February 2021) developed by the Andalusian Institute of Historic Heritage (IAPH)—the organization in charge of the protection and care of the regional heritage in Andalusia—is addressed below.

This online guide provides structured semantically enriched data (with links to the organization thesaurus (this thesaurus can be downloaded as RDF (Resource Description Framework) graph: https://guiadigital.iaph.es/tesauro-patrimonio-historico-andalucia, accessed: 15 February 2021—and DBpedia). Moreover, it aims at improving the accessibility and reuse of the resources concerning cultural assets that are managed by the organization

to both experts and general public. The guide provides access to the database of the IAPH regarding cultural heritage elements (consisting of file records with descriptions, images, panoramas and videos), where users can search and explore the collections classified according to the type of heritage: movable, immovable, intangible and cultural landscapes. All items are georeferenced, and the users can visualize a base map to locate them, as well as see the spatial relationships among the different elements and between the heritage items and the physical and human geography. Indeed, five pairs of maps and feature Open Geospatial Consortium (OGC) services (i.e., WMS and WFS services) are provided through the Andalusian SDI node of each of the four heritage types mentioned plus another SDI regarding cultural routes.

The following screenshot (Figure 2) shows an example of record in the guide. As can be seen, the elements are identified and have extensive descriptions that also include the people who were involved in their construction, remodeling or restoration. Likewise, information about the level of protection, bibliographical references, links with other records in the guide and to the institutional repository (which is discussed more extensively later in this report) are also available.

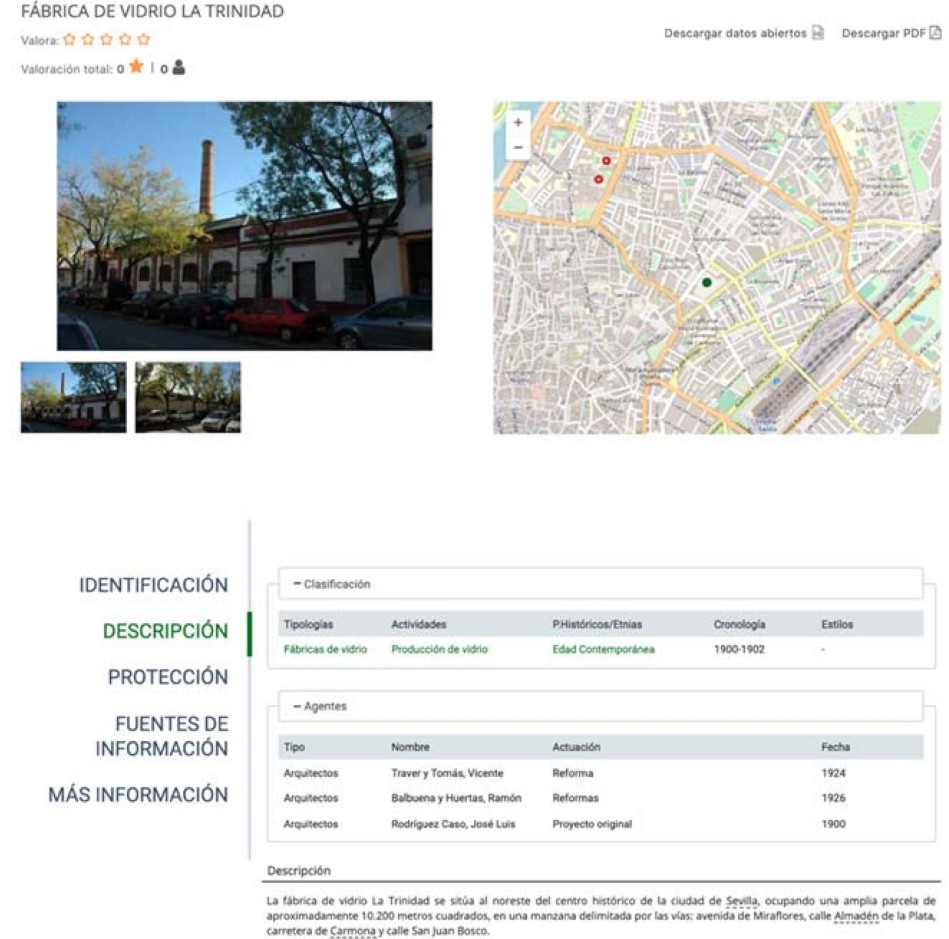

**Figure 2.** Record for the glass factory "Fábrica de la Trinidad". Apart from the identification of the element, a textual description, some images, the location on a map and links are provided. The records can be downloaded in PDF form for human readers and JSON (JavaScript Object Notation) for machine processing (https://guiadigital.iaph.es/bien/inmueble/8145/sevilla/sevilla/fabrica-de-vidrio-la-trinidad, accessed: 15 February 2021).

### 2.2. Repositories and Cultural Aggregators

Museums, archives and libraries have an extremely powerful resource in operation: Europeana, an aggregator for European cultural contents with a distributed structure

that allows for the participation of a wide range of agents in a coordinated way (similar examples can be also found in other geographical contexts, such as the Digital Public Library of America for the United States of America, Trove in Australia or DigitalNZ in New Zealand). In essence, it is a web portal in which searches can be run in order to find digital copies of cultural heritage items, but it does not digitize nor store those digital files; rather, Europeana manages metadata harvested from different online databases (the ones belonging to the cultural organizations) and supplies efficient tools for searching, browsing and linking data [28]. Indeed, during recent years, Europeana has been adopting the "linked data" model, which permits moving the focus from the documents to the elements of heritage, places, people, etc. by means of an encoding consistent with the "semantic web" ("semantic web" is a field of research focused on establishing efficient methods for data sharing, discovery, integration and reuse, which also includes the conceptual integration of heterogeneous sources and data [29]). This course of action also bridges the gap between different ways of information management within the many user communities that have created a set of systems based on manifold data structure standards. Examples are the ones used in museums, for example: Categories for the Description of Works of Art (CDWA) or Visual Resources Association (VRA) core categories; archives following the schemas proposed by the International Council on Archives, such as the International Standard Archival Description (ISAD(G)); or the libraries adapted to the standards issued by the International Federation of Library Associations and Institutions (IFLA). Likewise, it can cope with generic schemas (such as Dublin Core) and with more developed conceptual ones, such as the Lightweight Information Describing Objects (LIDO) [30] or the CIDOC-CRM (International Council of Museums—ICOM—International Committee for Documentation Conceptual Reference Model) [31], as well as make use of preferred terms, vocabularies, thesauri and, in general, any ontology.

In the wake of these steps, a number of research centers and universities—on the one hand—and cultural organizations such as ICOMOS (International Council on Monuments and Sites)—on the other hand—have started to gather, classify and give access to their datasets through networks of repositories. A pair of such examples are presented below; the former case concerns a small research structure (a university laboratory for the geometric documentation of heritage elements) which takes advantage of a pre-existing repository shared with other providers, whereas the latter shows the case of an organization which created its repository from scratch and for its exclusive use.

(a)    The following picture (Figure 3) shows a university repository which contains reports and results of projects for the geometric documentation of heritage elements [32]. There is no need to dwell on the importance of the information about the geometry and the appearance of the elements in order to track the evolution over time (photographs, line drawings, orthoimages, 3D models, etc.); however, when it comes to preparing the datasets for preservation, it is necessary to retain only the pieces of information that can be reused, store them in standard formats and describe the records according to the specifications of repositories and aggregators. Another concern is the process of rights clearance and the attribution of licenses for the reuse of the information that is provided, especially due to the complex web of relationships which appears when working with cultural heritage elements (artists, owners, custodians, providers, collaborators, etc.). Finally, in order to be considered by a cultural aggregator, the repository has to comply with a series of technical specifications regarding the accessibility, data quality standards and being in line with the scope;

(b)    In the same way, the picture below (Figure 4) shows the repository of the IAPH, which stores the information about the interventions on cultural heritage managed by this organization, technical documentation, information materials for public use, training materials, research reports and so on. The management strategy of this organization encompasses the entire life cycle of the documents from the very moment that the intervention concludes [33]. This approach not only guarantees that the information is properly preserved and available for reuse, but also contributes to

the generation of more reliable and authentic documents. At present, the repository particularly contains information on movable heritage, but it is expected that it will soon also include more information about other types (the information in the repository is continually growing; in order to have an up-to date view, see: https://repositorio.iaph.es/handle/11532/1, accessed: 15 February 2021).

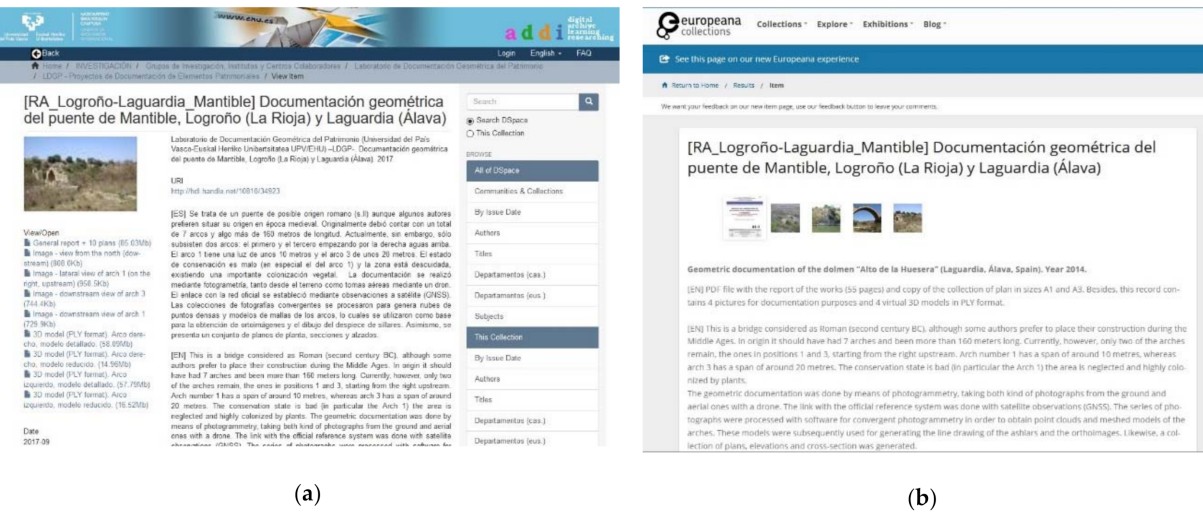

(**a**)       (**b**)

**Figure 3.** 3D documentation of a ruined bridge; the record includes the report describing the works carried out and characteristics of the attained results, a set of illustrative photographs and different versions of the 3D model. (**a**) Permanent record in the university repository (this record can be accessed through: http://hdl.handle.net/10810/34923, accessed: 15 February 2021); (**b**) the same item in Europeana.

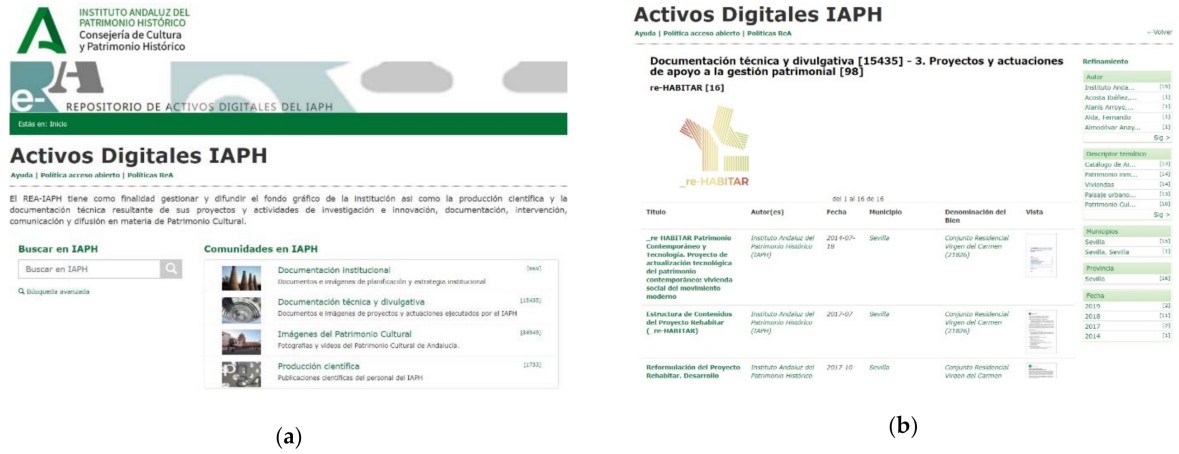

(**a**)       (**b**)

**Figure 4.** Two images of the Andalusian Institute of Historic Heritage (IAPH) repository (ReA). (**a**) Home screen of the repository, in which the contents are arranged in four sections: institutional documentation, technical and scientific documents, images of cultural heritage and scientific outcomes; (**b**) the repository as a dissemination portal of the research project entitled *re_Habitar*, in compliance with the requirement of the open access policy assumed by the organization.

### 2.3. Geographic and Three-Dimensional Databases

The improved capabilities of current databases and management systems stretch far beyond the specific applications previously mentioned and require further research. Indeed, a comprehensive view of the possibilities for the generation and management of knowledge makes it necessary to pay attention to several topics, among which it is worth mentioning the conceptual modeling of elements of heritage [34], which includes the elaboration of knowledge graphs and timelines for improving the understanding and visualization of the

elements of heritage and their relationships [35,36] and the formalization following standardized and machine-readable descriptions (e.g., RDF) for their implementation [37]. In addition, special mention should be made of the management via georeferenced databases (geographic information systems—GIS) and databases linked to 3D models (building information modeling—BIM) [38] and its application to cultural heritage building information models (HBIM) with a significant number of examples regarding restoration [39] and preventive conservation [40,41]. New conceptual and technological possibilities should be considered as a complement to the digital asset management (DAM) systems that, currently, are in use in many cultural organizations for the storage of multimedia files, data recovery, documentary analysis and so on [42].

In order to illustrate this, two different cases studies concerning the information management of historic buildings based on three-dimensional models—by means of heritage building information models (HBIM)—are presented as follows: the Charterhouse of Jerez [43] and the Renaissance Quadrant Façade of the Cathedral of Seville (Spain) [44]. In the context of the present paper, these particular cases were selected because they are two very different experiences in terms of the size of the buildings and objectives of the models: on the one hand, the Charterhouse of Jerez is a large monument complex in the surroundings of the city and, hence, the HBIM model was limited to the definition of the main lines of its master plan; by contrast, the Renaissance Quadrant Façade of the Cathedral of Seville is just a fragment of the building, but in turn, the model looked for high geometric accuracy and an exhaustive definition of its construction, by modeling every stone of the walls (Figure 5).

A key aspect in BIM is the level of development (LOD), which measures the geometrical and information complexity of a new building model (from a preliminary idea to the complete as-built definition). However, some interpretation is needed for historic buildings since, in these cases, the building already exists. A possibility is to seize this aspect to indicate the volume and reliability of the information available and the heritage management actions that can be supported by this information; thus, in this context, the concept of the level of knowledge (LOK) would be more appropriate, as this term suggests the need of a deep understanding of the heritage building prior to any action on it, in line with contemporary heritage intervention theories. The Charterhouse of Jerez HBIM model can be classified as level LOK200, which provides sufficient graphical accuracy and information to support actions related to legal protection and strategic planning. The need of documenting the existing knowledge about the Charterhouse as thoroughly as possible led to the establishment of a heritage information repository. The information was processed to obtain themed chronologies and timelines to support the HBIM modeling strategy. Furthermore, the heritage building itself was considered as a document because, even in the absence of an archaeological intervention, the pure observation of its walls and vaults in plain sight and the architectonical analysis showed the interpretation keys to generate the model.

Nevertheless, at present, such a large HBIM model is not functional if all the information is managed in a single model. Therefore, a base model and a series of partial independent models with different LOK types according to the existing information and historic relevance of each part and linked to the former were created. Hence, accurate models were generated where abundant information was available; in contrast, ruined and scarcely documented buildings were modeled with simplified shapes.

The Renaissance Quadrant Façade HBIM model reached the level LOK 400, i.e., it can provide support to specialized conservation works, with the stone ashlar being the basic constructive unit. The three-dimensional modeling started with a digital photogrammetric record of the façade, the generation of a point cloud and the associated mesh. This result was further processed in a CAD (Computer-aided design) environment in order to generate solid models according to three levels of geometric detail: low, medium and high. For the highest level, the model was cut into pieces (ashlars, balusters and rest of the constructive elements) taking as a reference the façade orthophotos obtained by photogrammetry.

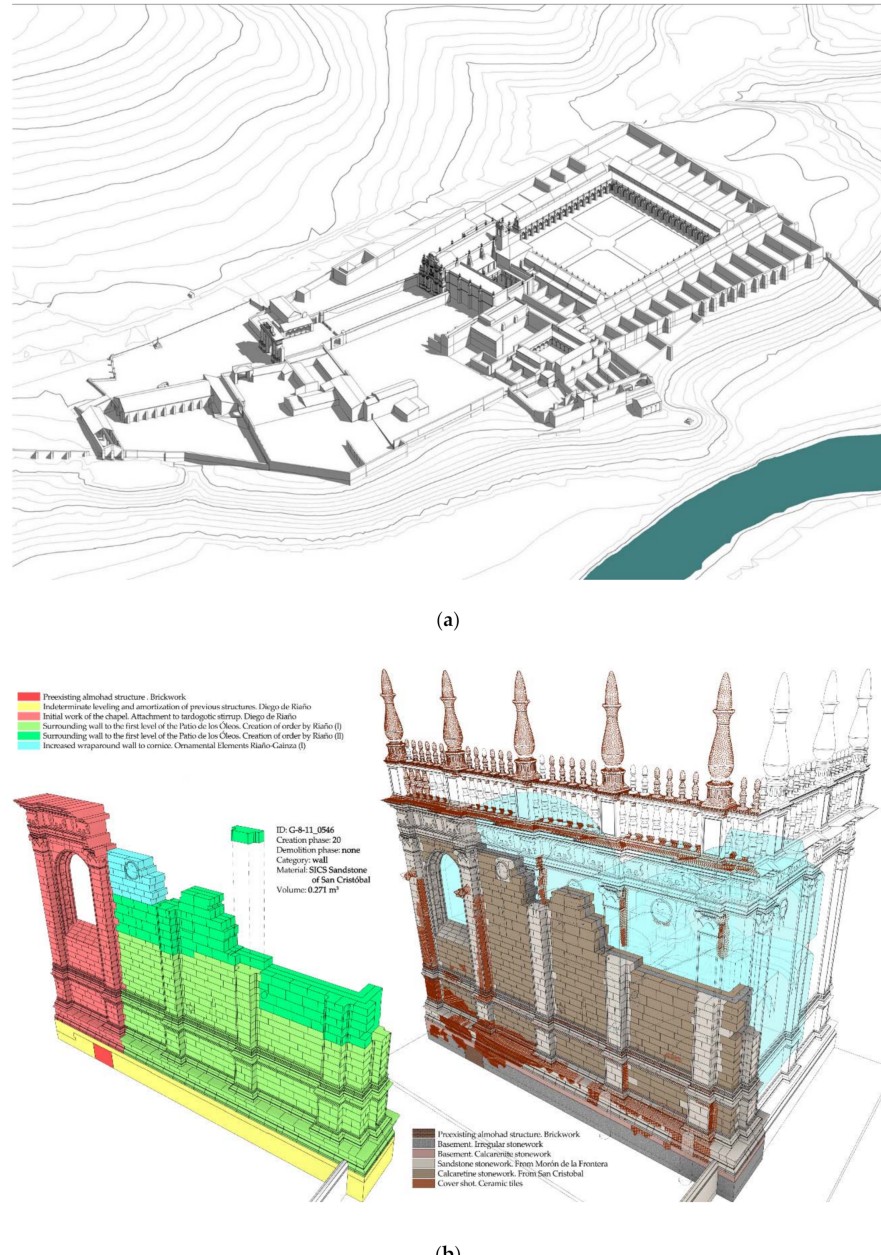

**Figure 5.** Examples of the employment of heritage building information models (HBIM): (**a**) HBIM model of the Charterhouse of Jerez, with a general view from the southwest corresponding to the last phase of the building evolution; (**b**) still image of the HBIM model of a fragment of the façade of the Renaissance Quadrant of the Seville Cathedral. On the left, there is a thematization of the model by the parameter "ID Phase", corresponding to its historical evolution, and the annotation of the parameters of one of its pieces. On the right, there is a thematization of the model by its materiality, including the interior spaces and a filtering of the surface information points (SIPs) by the parameter corresponding to the existence of biological deposits.

These CAD models were imported into the HBIM modeler software, categorized as walls and configured as information containers. The low- and medium-detail models registered the basic heritage information about the building as a whole, in a similar way to the Charterhouse of Jerez model. The highest level, however, includes the particular information relevant to its conservation: identification, belonging to a compositional element, chronology, damages, etc. Because damages on the surface of the stones (such as damp, deposits, alterations, material losses, etc.) are usually distributed without precise

correspondence with the ashlars, a new type of entity called the surface information point (SIP) was programmed. These SIP elements were distributed on the surface of the model following a net pattern that enabled their easy control and quantification.

## 3. Results

The following table (Table 1) provides an overview of the selected characteristics of the different approaches.

**Table 1.** Summary of the characteristics of the analyzed cases.

| | Open Data | | Repositories | | 3D Databases |
|---|---|---|---|---|---|
| | **SDI** | **Digital Guide** | **University** | **IAPH** | **HBIM** |
| 1. Information types | Map coverages | Texts, maps, images and thesaurus | Anything | Texts and images | 3D elements |
| 2. Information about the element of heritage or the surrounding environment | Environment (mainly) | Both | Both | Both | Element (mainly) |
| 3. Advanced built-in tools for data processing | Some geoprocessing tools | No | No | No | Yes |
| 4. Indexation tools | OGC Catalogue Services | OGC Catalogue Services and Aggregators (OAI-PMH) | Aggregators (OAI-PMH) | Aggregators (OAI-PMH) | Not yet available |
| 5. Online and interoperable accessibility | Yes | Yes | Yes | Yes | No |
| 6. Information for experts or for general public | Experts | Both | Both | Both | Experts |
| 7. Management transferred to an external permanent organization | Yes | No | Yes | No | No |
| 8. Development costs | Low | Medium | Low | Medium | High |

Concerning the types and scope of the information, the HBIM may be a very good starting point, not only due to the fact that the information is about the elements of heritage—something that is shared with the repositories—but also because this information is connected to a three-dimensional model which defines an architectural semantic of the building; consequently, it can be processed and presented in a variety of ways (e.g., a themed selection of elements with a definite chronology and state of repair, presented in form of an elevation plan) thanks to the advanced built-in tools. Conversely, the SDI are more suitable for gathering information about the surrounding elements and factors; likewise, they have the advantage that different coverages can be merged and analyzed in order to select elements within a delimited area, height, orientation and so on. For their part, repositories permit the inclusion of any type of information (texts, images, videos, etc.) stored in a large diversity of file formats and sizes (although, often, these files cannot be visualized directly online, i.e., they need to be downloaded and used locally).

Closely related to the types of information are the file formats used to store, disseminate and use it at different stages. In line with the open archival information system (OAIS) reference model (ISO 14721:2012 space data and information transfer systems), the definition of the file formats needs to be encompassed in the broader concept of the information packages, i.e., the logical containers that bind the information about the reference, provenance, content, context, representation and packaging. Moreover, there may be a difference between the package used for the internal management, such as the archival information package (AIP), and the options provided to the users, i.e., the dissemination information package (DIP).

Unlike the HBIM, all other cases analyzed were prepared to be remotely accessed following standardized protocols via specific indexation tools (catalogue services for the SDI and aggregators for the networks of repositories); this feature is closely related with the

following one (the online accessibility). Nevertheless, catalogue services and aggregators are two systems which operate separately, and they are not interrelated (i.e., an aggregator cannot search in a catalogue or vice versa). In principle, a combination which makes use of the semantic data concerning the element of heritage (stored in the repositories) together with the geographical information (in the SDI) would be possible, for instance, by extending the CIDOC-CRM to include spatiotemporal reasoning [45], but such an approach has not been implemented yet.

Output possibilities are certainly conditioned by both the purpose of the information (technical work, dissemination, research, etc.) and the user profile (architects, restorers, tourists, etc.). HBIM and SDI are particularly aimed at professional users with a wide knowledge about the system and the information stored therein. In line with the possibilities offered by the advanced built-in tools for data processing that are available in both approaches, the repositories are easily accessible to any user.

The second-to-last item in the table highlights that some of the alternatives take advantage of existing systems that are already operational by third parties, since they are in charge of public administrations (the case of the SDI) and universities (institutional repositories), among others (here we make a distinction between both repositories: On the one hand, there is the pre-existing university repository, which is managed by the library service. In this case, the research group that produces the information does not decide on how it works (neither will the maintenance entail any cost for it). In contrast, the IAPH repository was created specifically to store the documentation generated by the organization itself and is defrayed by the budged of the organization). Although the creators of the information lose, to some extent, the control over the datasets, this may sometimes be a favorable feature because it allows for the incorporation of systems administrators' expertise, the reduction of the workload (since most of the protocols and decisions to be taken are already defined) and counting on a group of users from the outset.

As for the expenses, the two approaches based on transferring the information to permanent organizations (i.e., the SDI and the university repository) avoid the costs of developing their own systems; in addition, they are very flexible over the acceptable type of information and formats with no expenses incurred by the providers, which qualifies both options economically. Conversely, the generation of a HBIM is labor-intensive and needs ongoing maintenance. Finally, both systems developed by the IAPH (digital guide and institutional repository) are indicated as medium cost because the organization needs to create and maintain them, though they are not necessarily very expensive, because they are based on standard protocols and the hardware and software requirements are moderate.

## 4. Discussion

In view of the results presented, a beneficial combination of the three technological systems—the HBIM for the detailed description of the heritage element, the SDI for the environmental factors and the repositories for the complementary information—should be developed.

However, maintaining three independent systems that need to be interconnected manually, since there is not yet any comprehensive solution, might sound overwhelming, even under optimal circumstances, which is not the case because each one of the components still has problems to overcome. Indeed, regardless the progress provided by the analyzed approaches, there are still some limitations and threats that impede their full development. For instance, the "open data" applied to cultural heritage have yet to become entrenched and still have some challenges to solve, both regarding the technological implementation and due to the reluctance of some administrations to adopt this policy [46,47]. An analysis that dates back to the initial steps of the "open data" implementation (2011) [48] showed the distinction of primary motivations to publish government data depending on the country (e.g., increase democratic control, foster service provision and strengthen law enforcement). Besides, the diversity of jobs involved in the conservation practice has generated different ways of managing information and, thus, a lack of interoperability, as well as situations

in which the existence of the information is unknown or the information is scattered [49]. Other issues, such as duplicated works, the disregard of already generated information or the non-recognition of contributors, entail the impossibility of merging information for different sources [50], and therefore, the coordination of different professionals. The "linked data" model and the application programming interface*s* (API) that are already included in the diverse data models may help troubleshoot some of the connectivity problems, but their development in the different cultural organizations is uneven and significant work is still necessary in order to make the datasets accessible according to these approaches [51]. Indeed, most of the institutional repositories of cultural organizations are still in the deployment stage [52,53]; even Europeana has an open debate about its sustainability and there are many cultural organizations that are on the outside of this or any other similar initiative (during the period 2014–2020, Europeana was funded through the tool Connecting Europe Facility (CEF), which was one of the most ambitious programs of the EU, covering issues such as cybersecurity or the transfer of health information; for the next period (2021–2027), the new program titled Digital Europe will take over [54]).

The set of case studies presented here enables the definition of some guidelines for effective information management concerning the conservation-restauration works. First, it is preferable to think in networks of interoperable systems rather than try to generate an all-embracing and centralized one; this way, each organization can be in charge of the maintenance of its own datasets and the community can support the development of the common tools and standards. As seen from the above, there are a large number of sources (e.g., SDI, repositories) that already provide useful information for conservation purposes, so they just need to be linked to make them available for further uses.

Some of the solutions presented, such as the HBIM, have a strong technological background. This is a double-edged sword: on the one hand, it is beneficial because they provide advanced functionalities but, on the other hand, they are very exposed to obsolescence and might need high investments in order to continue to be operational. Sometimes simpler and more cost-effective alternatives prove to be the more sensible options. In the case that the proposal is too specific and expensive, it may only be affordable for a reduced group of heritage elements and, consequently, the advantages of replicating the same solution on a large set of elements will not be obtained. Another important point is that the people who will use these systems (the ones in charge of the plans for preventive conservation) may be unfamiliar with these technologies, and thus, reluctant to employ them.

HBIM systems are seen as the confluence of two lines of development, which had several pioneering examples since the late 1990s: the idea of connecting the elements that make the 3D models up [55] to external databases, and the use of these 3D models to generate structured knowledge regarding the past building processes and the theory of architecture [56]. Thereafter, HBIM evolved to become a common space for gathering expertise about many disciplines (archaeology, architecture, history, restoration, etc.) [57] and a powerful tool for the generation of dissemination products [58], as well as for pooling information on sensors (e.g., weather conditions, movements, visits) installed all over the monument in the context of the monitoring necessary for preventive conservation [59,60]. Likewise, it allows for conducting scientific studies and simulations over the model, in line with the concept of a "digital twin" [61]. Nevertheless, though they can be efficient tools, there are some challenges regarding the use of HBIM for preventive conservation; the following should be noted in particular:

- The HBIM models are usually employed to show hypotheses about the original appearance of the buildings (e.g., [62]). However, this is not the pursued functionality here since, for conservation purposes, it is preferable to show the actual situation [63,64];
- The 3D modeling implies the formal definition of a semantic about the building process and this work requires an important amount of time and resources, whether for a HBIM generated from scratch or from unstructured 3D information (such as point clouds); there are some attempts to alleviate the workload by means of tools for

the semiautomatic generation, but they are still in their initial stages [65]. Therefore, HBIM systems can be time-consuming and expensive solutions, which also means that they cannot be applied to every case;

- Many authors have proposed the maximum geometric accuracy as the fundamental basis for the development of the HBIM systems [66,67]. This consideration, however, may disregard one interesting feature of the BIM technology: the possibility of progressively improving the quality of the models thanks to the definition of the levels of development (LOD), or better yet, the levels of knowledge (LOK), which allows for the adaptation of the model to the needs of each case, creator and end user. For instance, the two HBIM models presented in this paper showed accentuated contrasts in scale, purpose, geometric accuracy and information content; thus, they were developed in different levels of knowledge. Nevertheless, a LOK200 model like the Charterhouse one may gradually evolve to a higher LOK; symmetrically, a LOK400 model of a fragment of a heritage building, like the Renaissance Quadrant Façade, can be linked to a broader model in a lower LOK in order to contextualize the information stored in it;

- On the other hand, the architectural model is not the only type of relevant information for conservation purposes; therefore, the establishment of connections with other sources is also necessary, and this cannot always be done inside the HBIM. Thus, it might be interesting to export the structured knowledge outside the BIM to contextual (semantic) environments such as CIDOC-CRM [68,69], ConML (Conceptual Modeling Language) [70] or another specific domain ontology [71]. In addition, the connection among different HBIM systems remains to be solved; to this end, several options have been suggested such as the creation of networks of nodes [72] or solutions via the exportation to external formats such as CityGML (an application schema for the Geography Markup Language) [73,74];

- Finally, the operation of the HBIM requires specialized training and equipment (hardware and software) that is out of reach of most potential users. In this regard, more intuitive complementary tools have been proposed to facilitate management by non-expert users, such as interactive 360° panoramic views—coming from real image, virtual models or mixed reality—where information can be presented by means of hot points superimposed over the images [75].

Apart from the simple accessibility to the data, when it comes to information reuse, attention must be paid to the licensing system. In general, the less restrictive options usually are the ones which require less maintenance and allow for easier reuse; therefore, they are the most sustainable and productive ones in the medium term [76]. Likewise, although rights clearance may be a difficult matter [77], it is undeniably an essential part of the reuse and, consequently, it cannot be ignored.

Another important issue is the evaluation of the use, both quantitatively (how many users visualize and download the datasets) and qualitatively (what kind of uses arise). For instance, the next chart (Figure 6) shows the number of accesses during the second half of January 2021 to the record about the ruined bridge of Mantible, which is stored in the university repository (the one in Figure 3). A look at the statistics of use of the previous six months shows an average number of 6–7 accesses per month. The number of users that reach the repository are very far from the order of magnitude of the information distributed through social networking sites; however, that fact does not mean that the contents are unattractive for the general public (good photographs, for instance, are always well received, as well as the 3D models, videos, etc.), nor does it necessarily imply low-impact uses. In fact, the chart shows two peaks of use that are related to, on the one hand, the use of this record in an academic activity and, on the other hand, the interest aroused by the loss of a significant part of this element of heritage due to its partial collapse. Both situations prove the close cause-effect relationship between the events and the use of the information.

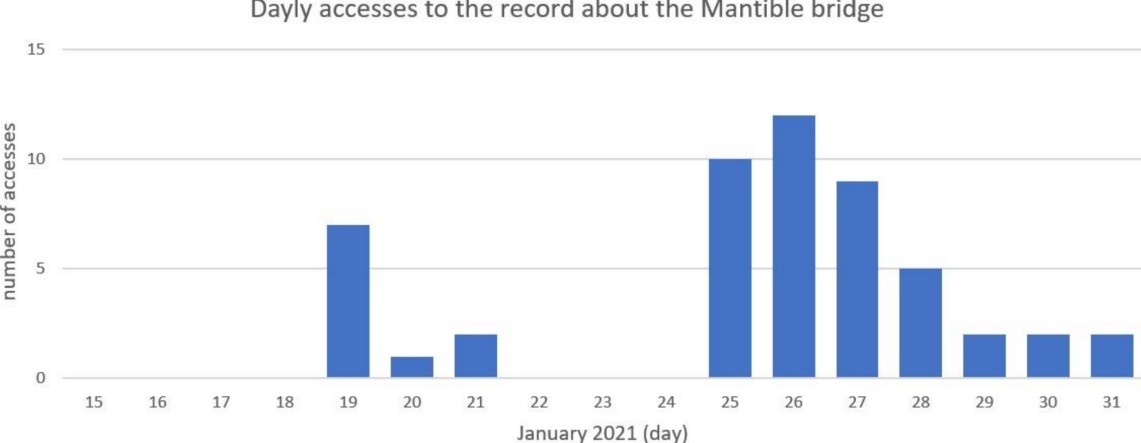

**Figure 6.** Accesses to the record with the 3D models, photographs, plans and report of the remains of the bridge of Mantible that are stored in the university repository. The group of accesses between 19 and 21 January 2021 were due to the employment of this record in the context of a lesson in a master course regarding cultural heritage management (i.e., the users were scholars). On the other hand, on the night of 24 January 2021, the ruined bridge collapsed, causing a huge upheaval among local citizens, and then, in the subsequent days, the use of the record reached a peak, mainly because of journalists who were looking for detailed technical information about the previous situation of the bridge in order to write their articles.

Concerning the sustainability and the medium- and long-term maintenance of the products, some of the analyzed case studies showed how the responsibility can be transferred to permanent organizations; this was the case of the SDI (transferred to the mapping service of the regional government) and the university repository (in charge of the university library service). This may be an option to consider when the information managers lack stable funding or there is no commitment for the upkeep of the generated assets inside the organization.

Likewise, once the issue related to the preservation and accessibility of the datasets have been solved, it will be time to think about the way of promoting the reuse of the information. To this end, it may be appropriate to establish links from customary resources (e.g., social networks, commercial platforms) known by the targeted users [78].

Summing up, there are many tools and systems that organizations in charge of the conservation of cultural heritage can implement. By way of recommendations, it can be said that:

(a)  If interoperability and immediate availability of the information for remote users are sought, the best option is to become a provider to an already existent environment, such as the network of repositories that are connected through aggregators or the SDI. Sometimes, the organization may have to set up the systems from scratch, which entails, on the one hand, the acquisition and maintenance of hardware and specific software and, on the other hand, staff training. This was the case for the two systems established by the IAPH (digital guide and repository). In other cases, the information is transferred to organizations which already have operational systems. This second option was the selected one in the examples of the SDI (that are managed by the regional mapping services of La Rioja and Andalusia, respectively) and the university repository. The differences are not just restricted to the costs and the time frame, because they may also affect the control over datasets and the technological expertise that the organization will gain as a result of the implementation. Consequently, each organization should evaluate its situation with respect to these aspects;

(b)  For the internal use of the information, a suitable digital asset management (DAM) system must be chosen. This may only include basic functionalities if the resources (financial and technical) of the organization are limited or it may take advantage of some improved capabilities. Certainly, these capabilities enable new uses, which

might justify the choice, as were the cases of the HBIM models presented in the text, but the initial costs are important either way;

(c)   It ought to be noted that, currently, there is no consistent tool for comprehensive management of both the internal and remote uses that encompass the historical documentation, monitoring, technical work and dissemination. In the meantime, the organizations need to develop different information systems in parallel, which involves a considerable workload, which should not be underestimated at the time of a new system implementation, for selecting, transforming and reprocessing the information.

Looking ahead in the evolution of the information management for the sustainable conservation of cultural heritage, some indications can be gleaned from the current research topics in the digital humanities. A non-exhaustive list of emerging topics that require more research may include the following aspects:

- It may be interesting to note the shift in attention from the data publishing to the development of tools for data analysis which, eventually, are expected to work autonomously [79];
- The challenge of integrating heterogeneous sources of cultural heritage knowledge will remain a field of joint study for computer scientists and domain specialists [80];
- The interconnection with other technologies, such as natural language processing (both for the content analysis of the documents and for the processing of the queries) or machine learning (with interesting applications in dealing with uncertainty, incomplete and fuzzy data, which are common characteristics of the information in the cultural heritage field);
- The optimization of interfaces and work processes is important, as many tools and software currently in use (such as Revit, Gephi, QGIS, etc.) require extensive data conversion into accepted formats. This entails huge efforts in terms of training on the use of each software to generate visualizations, models and analyses;
- Finally, it must not be forgotten that the ultimate aim of the information is to improve the management of conservation works, and thus, the maintenance and use of the heritage. Usage and impact monitoring are essential in order to justify the investments, which will entail the participation of actors from multiple areas such as economics, sociology, education or law.

## 5. Conclusions

At present, the conservation plan is the most appropriate instrument when it comes to tackling the generation, storage and use of the information regarding conservation and restoration works. On the one hand, this is because the plan is created and managed by an organization which has needs, responsibilities and resources to safeguard the cultural heritage. On the other hand, within the scope of the conservation plan, the information has a fundamental purpose, which is essential to decide whether it has to be generated or preserved.

To effectively manage monuments, a method is needed to rapidly assess the level of damage and vulnerability and to set the conservation priorities. The summary table presented in the Results section shows eight features, which are related with the decisions that are considered when envisaging a management system for conservation works (kind of information stored, expected users, maintenance costs, interoperability and sustainability of the system, etc.). In order to decide among the existing options and their possible combinations, different applicable approaches and technologies for information management were reviewed in this paper. In particular, the main approaches are summarized below:

- Data provided by the public administrations according to the "open data" policy, including the SDI; that is to say, information delivered by means of map layers;
- Repositories, for the distributed storage of any kind of information;
- HBIM, i.e., a detailed digital replica of the element of heritage considering its geometry structured on the basis of the architectural semantic, where each element can be connected to many thematic datasets.

They are all interesting for the development and support of the conservation plans. However, they are still not completely integrated for distributed and interoperable work.

Nevertheless, interoperability is highly desirable because it would entail benefits in the efficiency and sustainability of the management and preservation of cultural heritage. For this reason, this is an active research field, which is not limited to the availability, dissemination and processing of datasets, but also involves analyses of the associated rights and studies concerning the economic and social returns.

**Author Contributions:** Conceptualization, Á.R.M.; original draft preparation and writing, J.K.B. and Á.R.M. with the collaboration of A.Z.-I. (Introduction), J.M.V.-M. (case studies: SDI and university repository), P.A.I. and P.F.-L. (case studies: digital guide and repository of IAPH) and F.P.-P., M.C.-R. and R.A.-F. (case studies with HBIM). All authors have been involved in the creation of the comparative table in the "Results" section and have participated in the subsequent sections devoted to the "discussion" and the "conclusions". All authors have read and agreed to the published version of the manuscript.

**Funding:** The HBIM experiences presented constitute some of the actions developed in the HAR2016-78113-R Project, supported by the Ministry of Science, Innovation and Universities of the Government of Spain, R & D & I Plan, with the acronym TUSOSMOD. Two of the case studies mentioned in this text were created thanks to funding from the Instituto de Estudios Riojanos (Government of La Rioja), in particular: the SDI of wild grapevines, which presents a case study for open data (Figure 1), by means of a research grant in 2015; the documentation of the Mantible bridge, which illustrates the example of the university repository (Figures 3 and 6), as part of the public call "Planes 2017". The participation of J.K.B. in this research is supported by the Basque Government through grants for doctoral studies of the call 2019–2020.

**Institutional Review Board Statement:** Not applicable.

**Informed Consent Statement:** Not applicable.

**Data Availability Statement:** Not applicable.

**Acknowledgments:** The original text was significantly improved thanks to the assistance of the proofreading service of the campus of Álava (UPV/EHU). Likewise, we would like to acknowledge the (extensive) reviewers' work, who have substantially help to clarify many passages of the text and solve misunderstandings.

**Conflicts of Interest:** The authors declare no conflict of interest.

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
