# Peer review of "The Role of Information Management for the Sustainable Conservation of Cultural Heritage"

_sustainability, doi:10.3390/su13084325_

Round 1

Reviewer 1 Report

The paper covers a useful and interesting topic. I found it well written and sufficiently structured, but there are aspects that need improvement.

The article lacks rationale for the choices of examples it brings for the three categories covered.
I find the scientific content too descriptive, though certainly interesting.
I therefore suggest that the content be expanded upon (e.g.
Indexation tools from the table in line 435) and provide more motivation for the choices made.

Author Response

Dear reviewer [1],

We deeply appreciate the review carried out on the text and the interesting notes and asked improvement. After reading all the reviews the text has been extensively revised in order to fulfill all the requirements.

Regarding the particular concerns pointed out by you:

  • More attention has been given to the justification of each one of the cases analyzed in order to provide a sequential structure, which follows the logical chain of a line of thought.
  • We agree that the scientific content is mainly descriptive; however, this was a point of view that we missed in the existing literature regarding information management, which is mainly focused on describing specific proposals but lacks of comparative studies.
  • We have expanded several sections, especially, the results and discussion. The conclusions have been revised as well.

Moreover, the new version also includes changes indicated by other reviewers and as a result of the new reading done by the authors. These modifications include:

  • English language and style.
  • The section “Objectives” has been removed, and the information there was moved at the forefront of the introduction.
  • We have added some suggested bibliographical references and several new ones to address some gaps pointed out by the reviewers.
  • We have rearranged the subsection of “Materials and Methods” regarding “open data”. At present, both cases (SDI and digital guide) are together, which is more consistent with the other two subsections (repositories and 3D databases) and with the table in “Results”.
  • As indicated, the section regarding HBIM in “Materials and methods” has been renamed as: “Geographic and three-dimensional databases”.
  • We have rewritten some sentences in order to resolve the misunderstandings among concepts such as “open data” and “linked data”.

Yours sincerely,

The authors.

Reviewer 2 Report

This article is an interesting literature review that showcases the benefits of BIM technologies in the knowledge and data management of built heritage. Arguably, the strongest contribution is the analysis and showcase of the BIM technologies. However, there are still large gaps in the understanding on the role of ontologies and data models, and Semantic Web technologies in the management of built heritage. The authors present claims for Europe, basing themselves on very reduced literatures and understanding of data standards. Mos importantly it disregards the implementation of these socio-technical systems in the World – Wide Web -.

There are critical misunderstandings of the difference between Linked Data, Linked Open Data and Open Data. Especially since Open Data is a policy and not a technology. Further misunderstandings of the Semantic Web technologies are presented by the lack of awareness of the role of the Ontologies their knowledge representation and how through Linked Data and Semantic Web technologies the knowledge becomes interoperable.

Despite the aforementioned issues, I believe the strength of this article relies on the architectural technological and technical knowledge and how it can help better understand the lack of knowledge representation on the Web. I believe authors should consider downplaying the analysis of Semantic Web technology, briefly indicate the sustainable and disruptive elements of these technologies, and elaborate further how BIM data can merge with Content Management Systems (CMS), which are already interoperable through the ontologies, and then exploited through GIS systems (especially on the Web) through the aid of APIs.

Author Response

Dear reviewer [2],

Fists of all, thank you very much for the revision of the text, the notes and remarks provided were always pertinent and made us reflect about all the ideas pointed out. The text has gone through a comprehensive revision on the basis of the three reviews, that we have tried to combine in this new version.

Here are the main changes of the paper:

  • The section “Objectives” has been removed, and the information there was moved at the forefront of the introduction.
  • We have added to the text all the suggested bibliographical references.
  • We have rearranged the subsection of “Materials and Methods” regarding “open data”. At present, both cases (SDI and digital guide) are together, which is more consistent with the other two subsections (repositories and 3D databases) and with the table in “Results”.
  • As indicated, the section regarding HBIM in “Materials and methods” has been renamed as: “Geographic and three-dimensional databases”.
  • We have rewritten some sentences in order to resolve the misunderstandings among concepts such as “open data” and “linked data”.
  • On line 308 (previous version) there was a note warning us about the use of the description “multipurpose” for an ontology… actually, the word “multipurpose” was not about the ontology but about the Dublin Core metadata schema; in any case, we have deleted it.
  • The claim about the reluctance of some public administrations to share information (on line 492 of the previous document) is justified with two new references.
  • We have included the text suggested about the API’s in the note on line 500 (previous document).

However, we have preferred to maintain our original aim in the next two cases:

  • Regarding the two notes (on lines 64 and 67 of the previous version) regarding the traditional under-documentation of the restoration works and the resulting lack of awareness about its importance, the note indicated that this may be the situation in Europe but that a broader perspective should be considered. Indeed, it is true that this is the situation we know in our local context but the sentence is also backed —for a broader context— by the reference (Cruz et al., 2017) which makes these same remarks as part as the discussion after the international conference “Cultural Heritage: Disaster Prevention, Response and Recovery” –held in the end of 2016, in Lisboa, at the Calouste Gulbenkian Foundation.
  • On line 123 (previous version) there was a claim concerning the necessity of computer tools and legal basis for the correct information management. The remark indicates that these two are integrated in the data model (ontology). That may be true, however our initial idea here did not move to that direction but to advance the situation that is shown in the SDI, where it is a body of legislation (European Directives and regulations, national laws…) and a broad range of necessary technologies (file formats, communication protocols, software…). In addition, as one of the general recommendations of yours was to reduce the weight of the “semantics” in the text we have decided not to include the mention to the ontology in this paragraph.

Moreover, the new version also includes changes indicated by other reviewers and as a result of the new reading done by the authors. These modifications include:

  • English language and style.
  • Justification of the selected case studies.

Yours sincerely,

The authors.

Reviewer 3 Report

The paper explores a fascinating topic that needs to be discussed further in academic environments. It has the potential to be an interesting contribution to the field. Informations systems, while available, improving, and growing exponentially, are still lacking consistency and sustained support. The article identifies the necessity to have interoperability, outreach, and re-use of the different solutions. In that sense, the intentions are good, and the paper addresses the topic in a structured academic way.

However, there are several aspects that should be improved in this proposal:

  • While the building of the case is appropriate and it is well-complemented with the case studies and with sections expanding on the challenges of the different systems of conservation, I find the overall conclusions weak: they insist that more interoperability is necessary so that the conservation of cultural heritage can benefit from it. However, that conclusion lacks depth and concrete proposals on ways to provide that interoperability: it is good that the problem is identified, and you managed to build your case, but now what? Any concrete proposals? Who should be doing the integration? Who should be in charge of it? Who pays for it? Is there any successful example or platform integrating the different systems and can serve as a guide?
  • At times the article feels like a collage of data and information and cases. Although there is an effort to consolidate the findings in the Results (with table 1) and the discussion section, the results are far from providing a consolidated approach to the topic. More efforts in this direction need to be made. 
  • The language needs to be fine-tuned. The paper, despite proposing an interesting topic, is dense. That is not necessarily because the content is irrelevant or difficult to access, but because the long sentences and paragraphs make the reading experience particularly difficult (e.g. in page 4, there is a sentence that lasts from line 144 to 152). The writing style and how the content is presented need to be dramatically improved: this should be a condition for potential publication as the inconsistencies and, at times, unintelligible sentences diminish its potential to be understood. Often, I felt I was reading literal translations from another language, with numerous expressions that do not work in English, which forced me to constantly re-read the text to make sure I grasped the ideas. Please avoid expressions such as "anyway" (p.15, 506), "actually" (p.16, 537), "in any case" (p.18, 613), more suitable for a spoken language than academic work.

Author Response

Dear reviewer [3],

Thank you for your review, both the remarks and objections are most appreciated. We value very much the fact that you consider the topic worthwhile and that there is room for an interesting contribution on this field, we will address ourselves to this issue.  

The text has undergone an important revision with the aim of incorporate the comments and solve the criticisms. For instance, and with regard to some of the aspect point out by you:

  • Sections devoted to the “results”, “discussion” were expanded and more details are given about possible proposals and implementation guidelines. The “conclusions” have been revised as well.
  • We tried to justify better the rationale of each case study by itself and as a part of the complete set of examples considered in the context of the analysis carried out in the paper.
  • English language and style were revised.

On the other hand, further changes were also introduced both to comply with the comments expressed by the reviewers and as a consequence of the new reading given by the authors. In particular:

  • We have added some suggested bibliographical references and several new ones to address some gaps pointed out by the reviewers.
  • The section “Objectives” has been removed, and the information there was moved at the forefront of the introduction.
  • We have added to the text all the suggested bibliographical references.
  • We have rearranged the subsection of “Materials and Methods” regarding “open data”. At present, both cases (SDI and digital guide) are together, which is more consistent with the other two subsections (repositories and 3D databases) and with the table in “Results”.
  • As indicated, the section regarding HBIM in “Materials and methods” has been renamed as: “Geographic and three-dimensional databases”.

Yours sincerely,

The authors.

Round 2

Reviewer 2 Report

The paper has gone through substantial structural changes. There is still a limitation on the understanding of socio-technical systems when it comes to heritage management with Semantic Web technologies. However, being indicated that the main objective of the paper is to engage with the tools, and not necessarily the vocabularies used to support them. I believe that could be the following paper/extension for this research. 
I believe that due to the interdisciplinary nature of the paper, it is always healthy to downplay gaps in order to bring to light the main contribution. I think the paper is clear and well structured.

Author Response

Dear reviewer [2],

   Thank you once again for your intensive (and fruitful) examination of the paper. We also believe, as you have aptly pointed, that the subject of information management and the application for sustainable conservation of cultural heritage is vast and involves many disciplines and technologies (among other aspects that may be also considered), therefore, the attempt to offer a balanced contribution often conflicts with the necessity of going in depth in the analysis of all the different issues.

   We are pleased since, after the substantial restructuring of the paper, it seems that the text is ready for publishing. On a different note, we have noted down your recommendation to extend the research towards the understanding of socio-technical systems when it comes to heritage management with Semantic Web technologies.

   Some small changes concerning the footnotes and minor drafting changes have been included in this next version of the text.

Yours sincerely,

The authors.

Reviewer 3 Report

Thanks for addressing my suggestions. I recognize the effort you put into improving the paper. My three main concerns have been adequately addressed, and substantial enhancements have been implemented. I particularly appreciate the addition of the "summing up" section preceding the conclusion, which adds consistency and contextualizes better the individual cases discussed earlier in the paper; and the extensive work on the language, which makes the reading smoother.

Thus, I do not have further comments beyond a suggestion to review the way some footnotes (particularly in case study 2.1 and 2.2) are displayed (e.g. as links without adequate contextualization), and review the language again to polish minor inconsistencies (for instance, in line 674, the sentence is missing a "that")

Author Response

Dear reviewer [3],

    Thank you for your accurate remarks. We are glad to see that, after the changes, the resulting version of the work is more solid and, hopefully, interesting for the audience of the journal.

   As for the new corrections, we have contextualized the footnotes that only consisted of links (numbers 6, 10 and 11). And we have revised again the drafting and introduce around 15 small changes that we hope that will help to easy the reading and understanding of the text.

Yours sincerely,

The authors.